# Acne Syndromes and Mosaicism

**DOI:** 10.3390/biomedicines9111735

**Published:** 2021-11-21

**Authors:** Sumer Baroud, Jim Wu, Christos C. Zouboulis

**Affiliations:** 1Departments of Dermatology, Venereology, Allergology, and Immunology, Dessau Medical Center, Brandenburg Medical School Theodor Fontane and Faculty of Health Sciences Brandenburg, 06847 Dessau, Germany; sumer.baroud@gmail.com (S.B.); jimwuwang2@gmail.com (J.W.); 2College of Medicine, University of Sharjah, Sharjah P.O. Box 27272, United Arab Emirates

**Keywords:** acne, hormones, syndromes, endocrinology, immunology, autoinflammation, mosaicism, pilosebaceous unit, hirsutism, alopecia

## Abstract

Abnormal mosaicism is the coexistence of cells with at least two genotypes, by the time of birth, in an individual derived from a single zygote, which leads to a disease phenotype. Somatic mosaicism can be further categorized into segmental mosaicism and nonsegmental somatic mosaicism. Acne is a chronic illness characterized by inflammatory changes around and in the pilosebaceous units, commonly due to hormone- and inflammatory signaling-mediated factors. Several systemic disorders, such as congenital adrenal hyperplasia, polycystic ovarian syndrome, and seborrhoea-acne-hirsutism-androgenetic alopecia syndrome have classically been associated with acne. Autoinflammatory syndromes, including PAPA, PASH, PAPASH, PsAPASH, PsaPSASH, PASS, and SAPHO syndromes include acneiform lesions as a key manifestation. Mosaic germline mutations in the *FGFR2* gene have been associated with Apert syndrome and nevus comedonicus, two illnesses that are accompanied by acneiform lesions. In this review, we summarize the concept of cutaneous mosaicism and elaborate on acne syndromes, as well as acneiform mosaicism.

## 1. Introduction

The general public and many non-specialized physicians consider acne a mild disease, which resolves spontaneously in a matter of months to few years. On the other hand, its polymorphic clinical appearance, varying degrees of severity, acute as well as chronic forms and numerous subtypes lead to a confusion of acne vulgaris with a group of acne-like diseases, summarized under the term “acneiform dermatoses”. At last, there are complex cases presenting as part of syndromes, secondary to hormonal disturbances and/or genotype alterations, which sometimes makes the distinction of the clinical picture and its etiology in a certain patient quite difficult. The latter group, namely syndromic acne, is markedly under-recognized among physicians, and this makes this actualized review useful to clinicians and researchers.

## 2. Mosaicism

Mosaicism has traditionally been defined as the abnormal co-occurrence of at least two genotypes in a living being derived from a single zygote. However, recent studies have proven that mosaicism also occurs naturally as a physiological phenomenon [1], mainly due to a strikingly high but normal postzygotic mutation rate in utero, and variable somatic mutations throughout life [2]. Therefore, a more accurate definition of abnormal mosaicism is the co-occurrence of cells with at least two genotypes, present at birth, in a living being derived from a single zygote, which leads to a disease phenotype [3].

Mosaicism has frequently been classified in clinical genetics textbooks into three categories based on inheritance potential, namely somatic only, gonadal only, and both somatic and gonadal. Other classifications of mosaicism include two major subtypes, genomic and functional. Mosaicism, which is genomic in nature, is caused by a de novo change in the DNA sequence, giving rise to a genetically heterogeneous organism. Epigenetic mosaicism is the result of alterations which do not entail changes in the DNA sequence. This type of mosaicism is passed on from one cell generation to the next. The concept of mosaicism provides an explanation for why many mosaic disorders recognized in dermatology appear to be sporadic [1,3,4,5].

With genotypic diagnosis, many phenotypic clinical pictures are now being found to be part of a disease spectrum [6,7]. This spectrum is generally far greater in mosaic disorders than in germline disorders, simply because there are more factors that play role in affecting the ultimate phenotype. The terminal embryonic destination of the mosaic mutation and the onset of disease are somewhat linked, and, in general, earlier mutations will produce a more severe phenotype and will be more likely to be associated with non-cutaneous features. Other factors that alter phenotype include the exact mutation, with clear phenotype–genotype associations demonstrated in some conditions, the background germline genotype, and the normal function of the gene [8,9,10,11,12].

Somatic mosaicism can be further categorized into segmental mosaicism and non-segmental somatic mosaicism. The distinction between the two is particularly noticeable in cutaneous disorders. Skin lesions in non-segmental mosaic disorders may be diffusely disseminated and patchy with no respect for the midline, or they may occur at a single point [13]. Disseminated lesions most often result from postzygotic mutations. Examples include hereditary tumor syndromes: xeroderma pigmentosum, tuberous sclerosis, hereditary leiomyomas, and neurofibromatosis type 1 [14,15,16,17]. Patchy mosaicism not respecting the midline is rare and includes giant congenital melanocytic nevi. Finally, single point mosaicism includes all skin tumors not present due to a syndrome, such as most squamous cell carcinomas, basal cell carcinomas, and melanomas [18,19,20]. Moreover, due to the fact that melanocytes arise from neural crest cells, melanocytic mosaicism may involve other neural crest derivatives such as the peripheral nervous system, the craniofacial skeleton, and the smooth muscle cells of the aortic arch [21,22].

Segmental somatic mosaicism features cutaneous lesions that most often respect the midline and can follow a variety of patterns. The pattern depends on the type of cell that is affected and its trajectory of migration and proliferation during embryogenesis [23]. Segmental mosaics occur in five different forms of distribution: the checkerboard patterns, as in, e.g., melanosis neviformis Becker or nevus spilus, the phylloid type in mosaic trisomy 13, and the Blaschko line pattern in incontinentia pigmenti or the linear forms of autosomal dominantly inherited mosaics such as segmental neurofibromatosis type 1, Hailey–Hailey disease and epidermolytic ichthyosis [4,24,25,26,27].

## 3. Acne

Acne is a chronic illness [28,29] characterized by inflammatory changes around and in the pilosebaceous units, commonly due to hormone- and inflammatory signaling-mediated alterations of sebocyte differentiation, leading to changes in sebum quantity and release of lipid fractions, altered keratinization, and resulting bacterial colonization, such as with *Propionibacterium acnes* (*P. acnes*) and Staphylococci [30,31]. It most often affects hair follicles on the face, neck, chest, and back [32,33]. Acne’s prevalence peaks between age 14 and the start of the 3rd decade [33,34,35]. One population study in Germany found that 64% of those aged 20 to 29 years and 43% of those aged 30 to 39 years had visible acne [36]. The condition often persists into adulthood, with 26% of women and 12% of men reporting acne in their 40s [37]. About 60% of affected adolescents have mild acne, for which they use non-prescription preparations without consulting a physician. The remaining 40% constitute the population of acne patients seen in medical practice [29].

Risk factors for the development of acne include a family history of severe acne, the polycystic ovary syndrome (PCOS), the metabolic syndrome, and rare genetic conditions (e.g., Apert syndrome) [38,39]. The increased risk of developing acne in the presence of a positive family history ranges from 2.30 to 4.69 [40]. Familial clustering was evident in a retrospective study of 1557 pairs of monozygotic and dizygotic twins living in the UK [41]. The severity of acne, magnitude of sebum production and inflammation, extension of the disease, regional variations, clinical course, and response to treatment are influenced by genetic factors, such as family history of acne, as well as early onset of comedonal acne [32,42].

The formation of acne comedones in the skin depends on four processes: the release of inflammatory mediators such as interleukins, alteration of the keratinization process of the skin, androgen-mediated increase in sebum production and receptor hypersensitivity, as well as bacterial colonization of hair follicles by *P. acnes* [43]. *P. acnes* acts on the innate immune system through various proinflammatory pathways. The release of proinflammatory cytokines interleukin (IL)-12 and IL-8 is caused by the activation of Toll-like receptor (TLR) 2 on monocytes [44]. In addition, *P. acnes* prompts the release of IL-17A and interferon (IFN)-γ through the stimulation of CD4+ T cells [45]. Increased levels of IL-1 have also been reported in acne lesions. Factors that drive IL-1 expression include TLR2 induction by *P. acnes* and NF-κB-mediated transcription (stimulated by oxidized squalene). Thus, IL-1 might play a crucial role in the pathogenesis of acne, as it stimulates proliferation and keratinization of infundibular keratinocytes [46,47]. A higher level of CD4 cells and macrophages in uninvolved skin of patients suffering from acne vulgaris was reported, in comparison to the skin of patients without acne.

Follicular hypercornification through direct and indirect modulation of the innate immune system has been connected with changes in sebum lipid composition [48]. Sebaceous glands and sebum lipids are optimal anaerobic grounds for *P. acnes* activation [49]. Lipases produced by *P. acnes* hydrolyze triglycerides into pro-inflammatory free fatty acids as sebum passes through the follicular duct [50,51,52]. The binding of TLR2 and TLR4 on sebaceous glands by *P. acnes* triggers sebocyte production of antimicrobial peptides (human β-defensin (hBD)1 and hBD2) and inflammatory cytokines (tumor necrosis factor (TNF)α, IL-1α and IL-8) [44,53]. Moreover, various mechanisms of insulin-like growth factor (IGF)-1 may aid the emergence of acne, such as through increased androgen stimulation and the disinhibition of the forkhead box (Fox)O1 transcription factor, leading to further activation of the androgen receptor (AR). Although mild acne tends to settle down and eventually disappears in the majority of teen patients with acne, seborrhea often persists throughout adult life, long after inflammatory lesions have resolved [54]. Complications of acne can include physical symptoms such as soreness, itching, and pain [55]. Long-term complications such as scarring are more common in severe forms of acne. Acne’s emotional health effects include loss of self-esteem, lack of confidence, and symptoms of depression including suicidal thoughts [32,56,57]. The treatment of acne varies depending on the severity of the condition and can range from topical retinoids and topical antibiotics to oral antibiotics and isotretinoin [32,54]. Other forms of treatment include oral combined contraceptive medication for females with acne. Photodynamic therapy has also been proposed to manage inflammatory acne [58]. The depth and extent of acne scarring varies and can be improved by multiple procedures including subcision, punch excision, laser resurfacing, dermabrasion, and chemical peels [43,59]. Among these treatments, laser therapy seems to be the most effective to manage this aesthetic outcome [60].

## 4. Methodology of This Review

The present review is based on personal experience and a literature search in PubMed under the terms “acne”, “acne AND syndrome*”, “acne AND mosaicism”, “acne AND gene*”, “acne AND mutation*”, “acne AND hormone*”, “acne AND endocrine” and “acneiform”. References have been taken into consideration according to their relevance to the scope of the review.

## 5. Acne Mosaicism

Since family history of acne is a risk factor that predisposes an individual to develop the condition, genetics may play a pivotal role in the pathogenesis. Several theories have been proposed with regards to genetic causes, such as decreased CAG repeats in the *AR*, particularly in Asian individuals [61,62,63], polymorphisms in genes such as those for *TNFA*, *TNFR2*, *TLR2*, *IL1*, *CYP1A1*, *CYP17–34T/C*, *CYP21*, and somatic mutations in *FGFR2* [64,65,66,67,68,69,70,71,72].

Of the four known fibroblast growth factor receptors (FGFRs), i.e., FGFR1–4, the epidermis expresses FGFR1 and FGFR2. *FGFR2* is the gene encoding this family of related but individually distinct tyrosine kinase receptors. This receptor is alternatively spliced into two isoforms, FGFR2b and FGFR2c [73,74]. These have exquisite ligand specificity and are expressed in either epithelial (FGFR2b) or mesenchymal cells (FGFR2c) [74,75,76]. FGFR2b is an essential component in embryogenesis of the skin and is expressed by the epidermis, hair follicles, and sebaceous glands [77]. Its continued presence is necessary for the long-term survival of sebocytes, whereas its deletion leads to sebaceous gland atrophy [42]. *FGFR2* mosaicisms, which may be associated with epithelial dysregulation resulting in cancers of the breast, uterus, and skin, have been identified in a localized acneiform naevus in a single individual [64,78].

The germline mutations of adjacent amino acid residues of *FGFR2*, either *S252W* or *P253R*, cause Apert syndrome, an autosomal dominant condition characterized by severe acne and stereotypical craniofacial and limb abnormalities [64,79,80,81,82]. These gain-of-function mutations are localized in the linker region between D2- and D3-immunoglobulin-like regions of the FGFR2-ligand-binding domain [82]. The resulting mutation causes inappropriate receptor activation due to loss of ligand specificity for both the FGFR2b and FGFR2c isoforms, leading to follicular keratinocyte proliferation, sebaceous lipogenesis, and inflammatory cytokine response [82,83,84,85,86]. FGFR2b is exclusively expressed on epithelial cells, whereas FGFR2c is only expressed on dermal and mesenchymal cells [87]. In the *S252W* mutation, the mesenchymally expressed isoform FGFR2c binds more strongly with FGF7 and FGF10, while the epithelial isoform FGFR2b binds more with FGF2, FGF6, and FGF9. Increased signaling of mutated *FGFR2* upregulates the activity of phosphoinositol-3 kinase (PI3K)/Akt and MAP kinase signal transduction pathways [88]. With that being said, PI3K⁄Akt activity is not only increased by FGFs but also by other growth factors such as insulin and IGF-1, thus offering a plausible explanation why individuals with metabolic syndrome suffer a more severe onset of acne [86,88]. Moreover, the loss of delayed receptor internalization and lysosomal FGFR2 degradation plays a major role in the increased *FGFR2* signaling activity in Apert syndrome [89]. 

Studies have also reported that FGF synthesis is stimulated by the presence of androgens, which may explain why individuals with Apert syndrome develop an early onset of severe acne during puberty. Melnik et al. report that the production of FGF7 and FGF10 by fibroblasts is augmented in response to 5α-dihydrotestosterone via transcription. FGF7 and FGF10 are ligands for FGFR2b found on suprabasal keratinocytes and sebocytes [88]. FGFR2b binding leads to transcription of IL-1, which encourages hyperkeratinization, and in sebocytes, proliferation and fatty-acid synthesis [89,90]. The use of low dose isotretinoin and antiandrogens has been reported to be effective in treating acne in Apert syndrome, further validating the theory of androgen-dependent up-regulation of FGF synthesis [79,80,86,91,92,93].

## 6. Acne-Associated Endocrine/Immunological Syndromes

Systemic endocrine/immunological disorders and syndromes such as congenital adrenal hyperplasia (CAH), PCOS, seborrhea-acne-hirsutism-androgenetic alopecia (SAHA) syndrome, primary ovarian insufficiency, and many others have classically been associated with acne [87,94] (Table 1). These conditions should raise suspicion and considered as plausible diagnoses when patients present with sudden, severe, and/or treatment-resistant acne, both in childhood and adulthood. Moreover, acne that presents with other signs of hormonal imbalance, such as hirsutism, irregular menstruation, altered libido and insulin resistance, should raise clinical suspicion of an acne-associated syndrome [87,94].

### 6.1. CAH

CAH (OMIM 201910) consists of a heterogeneous group of autosomal inherited disorders due to enzymatic defects in the biosynthetic pathway of cortisol and/or aldosterone, resulting in hormonal imbalance [87,95]. *CYP21* (6p21.3) mutations result in 21-hydroxylase (21-OH) deficiency, which occurs in 95% of CAH cases [96]. Non-classical adrenal hyperplasia (NCAH) is the milder form of CAH, which often occurs later in life. Through the impaired cortisol-mediated negative feedback control of adrenocorticotropic hormone (ACTH) secretion from the anterior pituitary gland, cortisol deficiency in CAH causes oversecretion of ACTH and overstimulation and hyperplasia of the adrenals.

Cutaneous manifestations of NCAH, such as androgenic alopecia, hirsutism, and acne, are caused by androgen excess affecting the pilosebaceous unit [97]. NCAH treatment should address the main problem of the patient, such as acne or hirsutism (Figure 1). Oral glucocorticoids are administered in order to counteract adrenal androgen production for the treatment of acne connected with NCAH [97,98].

### 6.2. PCOS

One of the most typical endocrine disorders in females of reproductive age is PCOS (OMIM 184700). The accumulation of incompletely developed follicles in the ovaries result in irregular menses and other clinical signs of hyperandrogenism including acne vulgaris, seborrhea, hirsutism, infertility, and alopecia [99] (Figure 2). 

With serum-free testosterone being the most sensitive biochemical marker, increased circulating androgen levels are noticed in 60–80% of patients. Females who have severe and persistent acne vulgaris are reported in about 1/3 of women with PCOS [100]. Hyperandrogenemia, altered gonadotropin secretion, as well as insulin resistance are involved in PCOS pathogenesis.

### 6.3. SAHA Syndrome

The combination of seborrhea, acne, hirsutism, and androgenetic alopecia form SAHA syndrome, which was first described in 1982 [95,101] (Figure 3). Cutaneous manifestations may be caused by hyperandrogenemia or heightened sensitivity of the pilosebaceous unit to normal circulating androgen levels. As seen in disorders such as constitutional seborrheic skin in women, androgen excess can grow the size and number of lobules per sebaceous gland as well as sebum excretion [102,103]. It is theorized that androgens may play a permissive job in influencing acne development, or it may be the local overproduction of androgens in the skin that leads to acne [102]. The pattern of the syndrome can markedly vary regarding the expressiveness and the severity of the clinical signs from only cosmetical discomfort to serious illness [95,101,103] (Table 2). Lifestyle modification, oral contraceptives, anti-androgens, and insulin-sensitizing medications are treatments for the dermatological conditions of hyperandrogenism [104,105].

### 6.4. Primary Ovarian Insufficiency

Primary ovarian insufficiency (OMIM 311360), also known as premature menopause, premature ovarian failure, or early menopause, is defined as the occurrence of amenorrhea before the age of 40 accompanied by increased follicle-stimulating hormone to menopausal levels (>40 mIU/mL) and decreased estradiol levels (<50 pg/mL) [106,107]. It occurs in about 1% of the female population under the age of 40 years and is occasionally diagnosed after discontinuation of hormonal contraceptives [106]. Even though presenting with clinical findings similar with those of menopause, 50% of primary ovarian insufficiency patients have varying and unpredictable ovarian function, and only 5–10% are able to accomplish pregnancy. Estrogen deficiency directly affects skin homeostasis and may also result in dysregulation of peripheral androgen metabolism [108]. Seborrhea, acne tarda, and male-pattern baldness may likely be results of the relative hyperandrogenism occurring in primary ovarian insufficiency together with persisting amenorrhea [107] (Figure 4).

## 7. Acne in Autoinflammatory Syndromes

Autoinflammatory syndromes that involve acne include pyogenic arthritis-pyoderma gangrenosum-acne (PAPA) syndrome, pyoderma gangrenosum-acne-suppurative hidradenitis (PASH) syndrome, pyogenic arthritis-pyoderma gangrenosum-acne-suppurative hidradenitis (PAPASH) syndrome, psoriatic arthritis-pyoderma gangrenosum-acne-suppurative hidradenitis (PsAPASH) syndrome, pustular psoriasis-arthritis-pyoderma gangrenosum-synovitis-acne-suppurative hidradenitis (PsAPSASH) syndrome, pyoderma gangrenosum-acne-suppurative hidradenitis-ankylosing spondylitis (PASS) syndrome, and synovitis-acne-pustulosis-hyperostosis-osteitis (SAPHO) syndrome [87,109] (Table 1). The type of acne associated with autoinflammatory syndromes is usually a severe nodulocystic type, clinically presenting with large, inflamed nodules and cysts affecting the face, back, and chest [110].

The IL-1β pathway has been associated with many of the autoinflammatory disorder mutations [111]. The dysregulation of inflammasome function and the release of proinflammatory cytokines, such as TNFα and IFN-γ, and chemokines, particularly IL-8 and RANTES (regulated on activation, normal T cell expressed and secreted), which are responsible for the recruitment and activation of neutrophils that lead to neutrophil-mediated inflammation, is usually related to the over-expression of IL-1β [112,113]. Proinflammatory cytokines also induce the production of metalloproteinases (MMPs), notably MMP-2 and MMP-9, leading to tissue destruction by degrading the components of the extracellular matrix [114]. MMPs of epithelial origin have also been detected in facial sebum of patients with acne and they are regulated by isotretinoin treatment [115].

### 7.1. PAPA Syndrome

PAPA syndrome (OMIM 604416) combines the triad of pyoderma gangrenosum, acne, and pyogenic sterile arthritis, inherited in an autosomal dominant fashion. Genetic alteration of *PSTPIP1* (proline-serine-threonine phosphatase-interacting protein 1 gene) on chromosome 15 (15q24-q25.1) is the most commonly reported inciting cause [87,116]. Cutaneous involvement differs among patients. During puberty, dermatologic manifestations become more notable, usually premiering with tender acne lesions and subsequent development of painful papules or nodules that break down into a rapidly enlarging and persistent ulcer, usually located on the lower part of the body. Upon minimal trauma, pustule formation followed by ulceration may occur early in life. These cutaneous signs and symptoms usually persist into adulthood [117]. The diagnosis can be confirmed by identification of a mutation in *PSTPIP1*. For treatment of the clinical symptoms of PAPA, IL-1, and TNF blocking agents are usually successful, and thus the mainstay of management [112].

### 7.2. PASH Syndrome

PASH comprises the syndromic triad of pyoderma gangrenosum, acne, and hidradenitis suppurativa and was first described in 2012 [118]. Pathogenesis is primarily an increase in proinflammatory cytokines IL-1 and TNF in the skin of PASH patients, possibly due to an increased number of CCTG repeats in the *PSTPIP1* promoter region, a finding consistent PASH patients [118,119]. Various types of cutaneous lesions such as pyoderma gangrenosum-resembling ulcerated nodules, pustules, HS-like abscesses and fistulae, and facial acne have been reported [112,120]. Treatment options for PASH are limited due to the unclear pathogenesis. Infliximab as monotherapy has achieved conflicting results among patients, sometimes leading to remission and other times failing. Rapid improvement of all cutaneous symptoms and long-term remission in a severe case of PASH usually occurs with the combination of infliximab, cyclosporine, and dapsone [121].

### 7.3. PAPASH Syndrome

The concomitant diagnosis of pyogenic arthritis, pyoderma gangrenosum, acne, and hidradenitis suppurativa consists of a syndrome known as PAPASH. Genetic studies have revealed a missense mutation (p. E277D) in the *PSTPIP1* gene in patients suffering from PAPASH [122]. Treatment is closely related to the pathogenesis of the disease. The efficacy of IL-1 inhibition with anakinra or canakinumab has been well documented in several studies of autoinflammatory syndromes, including PAPASH, confirming the pro-inflammatory role of dysregulated *IL1* signaling [123,124].

### 7.4. PsAPASH Syndrome

The triad of pyoderma gangrenosum, acne, and hidradenitis suppurativa in combination with psoriatic arthritis represents the PsAPASH syndrome [125].

### 7.5. PsAPSASH Syndrome

A new syndromic phenotype comprising pustular psoriasis, arthritis, pyoderma gangrenosum, synovitis, acne and suppurative hidradenitis was named PsAPSAH [126]. The pathophysiology based on the dysregulation of IL-17 production provided the rationale for a successful treatment with the IL-17A inhibitor secukinumab.

### 7.6. PASS Syndrome

PASS syndrome is a rare inflammatory disease characterized by a chronic-relapsing course of pyoderma gangrenosum, acne, hidradenitis suppurativa, and ankylosing spondylitis. Patients display episodes of fever along with elevated serum levels of IL-1β. Skin lesions are characterized by sterile neutrophilic infiltrates and show a rapid response to the IL-1 receptor antagonist anakinra [127].

### 7.7. SAPHO Syndrome

The combination of synovitis, acne, pustulosis, hyperostosis and osteitis define the SAPHO syndrome [128], an uncommon, chronic inflammatory disease. The illness is characterized through the involvement of bone, joints, and skin, and typically presents with pain, swelling, and tenderness in affected areas [128,129] (Figure 5). Cutaneous manifestations, which are neutrophilic in nature, tend to develop well before or after the emergence of osteitis, the common denominator in the majority of SAPHO syndrome patients [129,130]. The skin is most commonly involved in the form of palmoplantar pustulosis, representing 50–75% of dermatologic manifestations [131,132]. Psoriasis has been reported in 1/3 of patients with skin involvement. Severe acne affects approximately a quarter of patients with SAPHO syndrome [131]. The pathogenesis of SAPHO syndrome is unclear; however, the impressive resolution of the illness in response to anakinra indicates that SAPHO syndrome may be associated with genetic variants of *IL1RA* or related genes [133,134].

## 8. Acneiform Eruptions

The manifestation of acne/acneiform lesions in numerous diseases/syndromes of varying etiology highlights the multifaceted nature of acne [28,29,87,94,102,124]. Since not all eruptions that resemble acne also represent acne vulgaris, awareness of diseases is essential, which can neither be classified under acne nor are part of a systemic disease or syndrome [135,136,137] (Table 3).

## 9. Acneiform Mosaicism

### 9.1. Apert Syndrome

Apert syndrome (OMIM 101200), also known as acrocephalosyndactyly, is characterized by synostosis of the distal extremities, the vertebral body and skull bones, along with syndactyly of the fingers and toes. It has a birth prevalence of 1 in 65,000 [138]. Although the majority of cases occurs sporadically, autosomal dominant transmission and germinal mosaicism have also been documented [82,86,87,139]. The condition was first described by Wheaton in 1894 and later by Apert in 1906 [140]. Apert syndrome was first described in dermatology in 1971 by Solomon after observing several patients who presented severe acneiform lesions in unusual distribution along the forearms [141] (Figure 6). Today, cutaneous manifestations are characteristic of Apert syndrome, such as hyperhidrosis, loss of eyebrow hair, forehead wrinkling, and skin dimpling over knuckles, shoulders, and elbows [142]. Due to progressive osseous fusion of the tarsal and metatarsal bones, weight bearing shifts to the mid- and lateral plantar regions in most patients, leading to lateral plantar hyperkeratosis. Patients may additionally display hypopigmentation of the skin and eyes due to failure of melanoblast migration in utero [143]. Cutaneous mosaicism for Apert syndrome mutations results in regional manifestations of acne along Blaschko lines, indicating that the mutations are directly responsible for the formation of acne lesions [64]. Seborrhea is noted at adolescence, with subsequent appearance of follicular acneiform papules, pustules, and comedones, diffusely involving the face, upper trunk, forearms, buttocks, and thighs [79,144]. Histology shows dilated infundibula with keratinous material, corresponding to comedones and enlarged sebaceous gland lobules [79].

### 9.2. Acneiform Unilateral Nevus—Nevus Comedonicus

Nevus comedonicus (OMIM 617025) is a rare (prevalence 1:45,000–1:100,000) epidermal nevus comprising of a group of dilated hair follicle openings filled with plugs of brownish-black oxidized keratin [145,146]. The age of onset is most prevalent in birth and childhood [147]. It mainly occurs unilaterally and is most often localized on the head and neck; it was initially described as “localized acne” [64] (Figure 7). If it occurs as a part of the nevus comedonicus syndrome (NCS) [148], it can be generalized and form linear streaks. Hidradenitis suppurativa-like lesions have been observed in naevus comedonicus [81]. Other cutaneous features include hypopigmentation and/or hypotrichosis of a nevus with less pigmented terminal hairs with abnormal curled growth pattern and reduced density, as compared with unaffected adjacent skin [87]. Unilateral acneiform nevus is considered a more severe form of nevus comedonicus [149]. Histology often shows dilated plugged follicular infundibula resembling comedones, with small pilosebaceous follicles and hypertrophic sebaceous glands, unusually found in the upper third of the dermis, as in Apert syndrome. A *Ser252Trp* missense mutation in *FGFR2* has been documented in some cases [64]. The first indication of the presence of an epidermal genetic mosaic due to *Ser252Trp-FGFR2* mutation in the form of acneiform nevus was documented in a 14-year-old teenage boy who presented with sharply bordered acneiform lesions along the lines of Blaschko on the left shoulder and elbow as far as the extensor aspect of the left forearm, with comedones in almost all follicles [64]. The detected mutation is identical to the *Ser252Trp* germline mutation of *FGFR2* in Apert syndrome, and as a result, the acne seen in Apert syndrome and unilateral acneiform nevus respond similarly to treatment. The response to oral tetracycline is inadequate, while oral isotretinoin effectively leads to improvement of the lesions [150,151].

### 9.3. Happle-Tinschert Syndrome

Happle–Tinschert syndrome (OMIM 109400), formerly categorized as basal cell nevus syndrome, is characterized by segmentally arranged basaloid follicular hamartomas organized in a systematized pattern along the lines of Blaschko, associated with extracutaneous defects of the bones, teeth, and brain [152]. According to Burck and Held [153] and Peter [152], a rather unique clinical feature of the skin lesions, present in early infancy, seems to be linear hypopigmentation. Histology of the lesions show basaloid follicular hamartomas, some of which may display a central comedone-like plug, a finding commonly found in naevus comedonicus syndrome [154]. Interestingly, increased in situ expression of melanocortin-1 receptor has been shown in sebaceous glands of lesional skin of acne patients [155].

### 9.4. Becker Naevus

Becker naevus (OMIM 604919) is a cutaneous hamartoma characterized by circumscribed hyperpigmentation with hypertrichosis [156]. Becker naevus syndrome refers to the association of Becker naevus with developmental defects such as ipsilateral breast or muscle hypoplasia and cutaneous abnormalities such as acneiform lesions [157,158]. Acne lesions confined to Becker naevus have been described in several cases, including asymmetrical and unilateral lesions, commonly involving the thorax [146]. The occurrence of two asymmetrical Becker naevi in the same person supports the cutaneous mosaicism (somatic) theory of origin of Becker naevus, characterized by the presence of two or more populations of genetically different cells derived from the same zygote. Becker naevus follows the type two clinical pattern (checkerboard pattern) of cutaneous mosaicism described by Happle, making it unique from other acneiform mosaic conditions [5]. Person and Longcope described a case of Becker naevus with acneiform lesions and noted increased AR levels in lesional tissue compared to normal skin, supporting the theory that heightened androgen sensitivity and stimulation may be precipitating the clinical manifestations of Becker naevus [159,160].

## 10. Conclusions

Acne represents a cardinal skin manifestation in several groups of diseases. It is present in endocrine/immunological syndromes, autoinflammatory syndromes, and certain mosaicisms, indicating that it is an intriguing model for studying the interactions among hormones, innate immunity, inflammation, wound healing (scarring), and gene mutations (Figure 8). Exploring possible common mechanisms of acne induction in various acne-associated diseases and syndromes—including mosaicisms—might contribute to the development of novel therapeutic regimens.

## Figures and Tables

**Figure 1 biomedicines-09-01735-f001:**
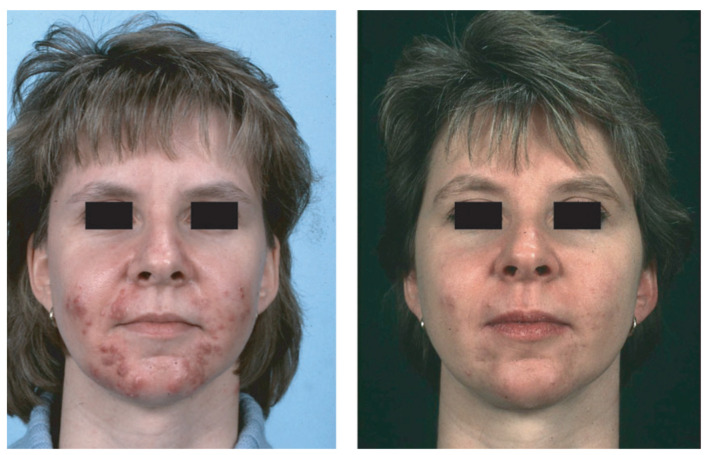
NCAH recalcitrant to hormonal antiandrogens and systemic isotretinoin—in a 31-year-old woman before (**left**) and after (**right**) 2 months of treatment with prednisolone 5 mg/d. Reproduced from Zouboulis and Piquero-Martin [98] with permission from S. Karger AG, Basel.

**Figure 2 biomedicines-09-01735-f002:**
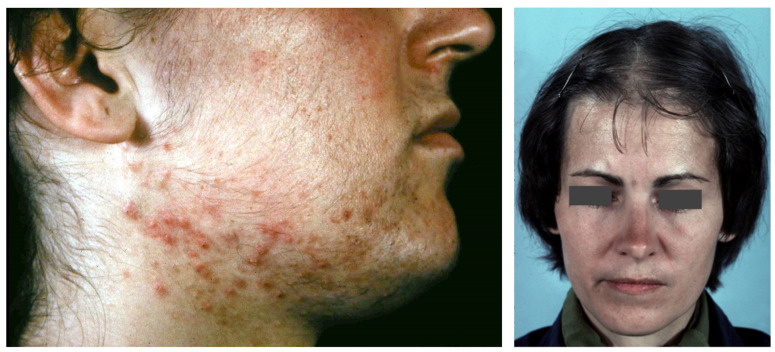
Acne, hirsutism and androgenetic alopecia of female pattern in PCOS patients.

**Figure 3 biomedicines-09-01735-f003:**
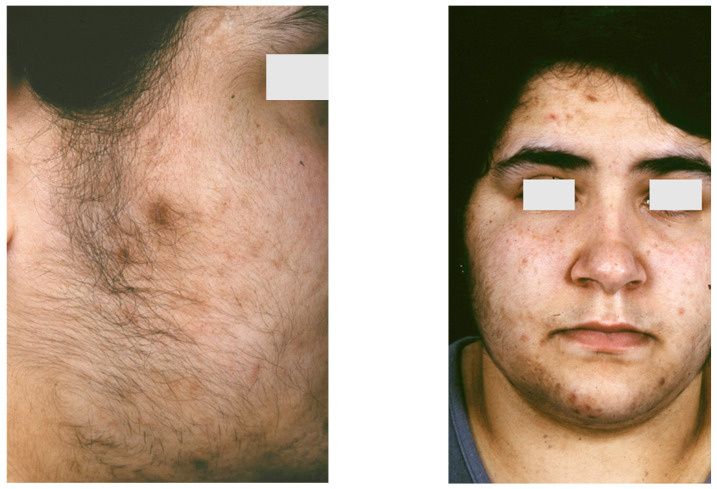
An 18-year-old female patient with SAHA syndrome (HAIRAN variant); seborrhoea, acne, and hirsutism are shown. Further clinical signs: Obesity since childhood, menarche with 11 years, menstruation every 2–3 months, insulin resistance. Right picture reproduced from Chen et al. [87] with permission from Wiley, Hoboken, NJ, USA.

**Figure 4 biomedicines-09-01735-f004:**
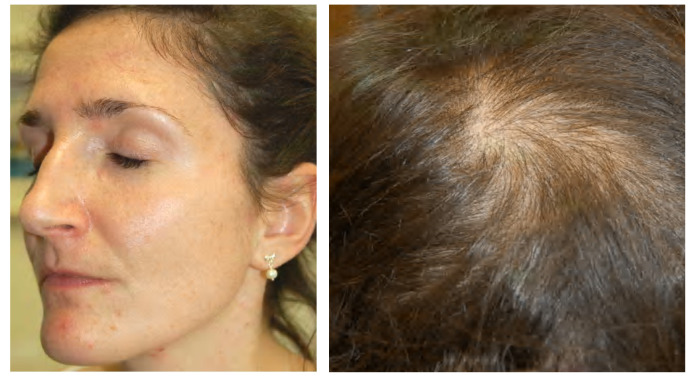
A 30-year-old female patient with primary ovarian insufficiency. Clinical signs: Amenorrhea, seborrhea, acne tarda and male-pattern baldness. Reproduced from Zouboulis et al. [107] with permission from S. Karger AG, Basel.

**Figure 5 biomedicines-09-01735-f005:**
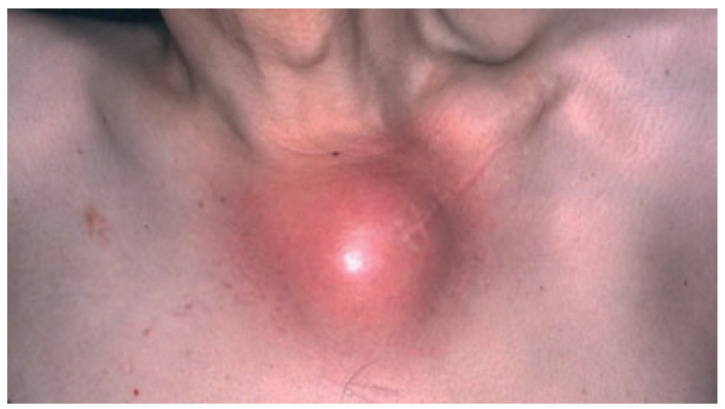
SAPHO syndrome showing involvement of the left sternocostoclavicular area with inflammation and swelling. Reproduced from Chen et al. [87] with permission from Wiley, Hoboken, NJ, USA.

**Figure 6 biomedicines-09-01735-f006:**
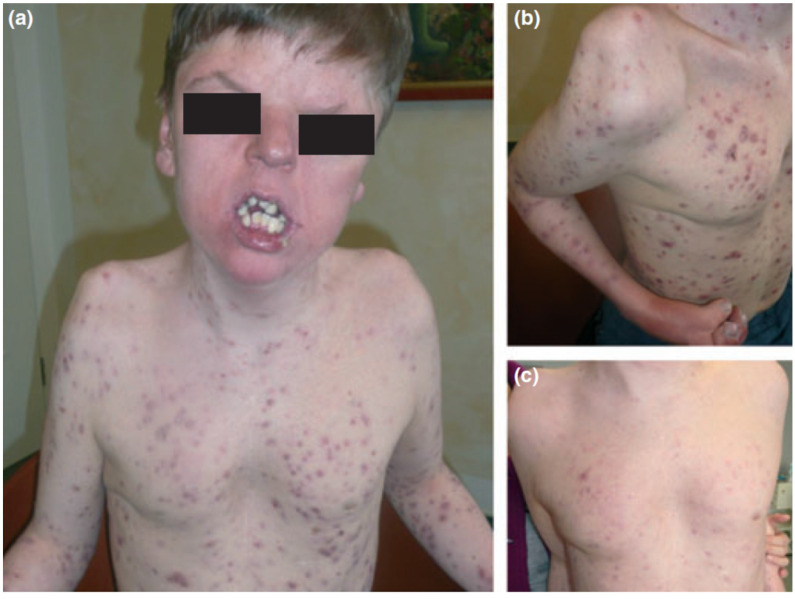
Apert syndrome with early onset severe nodulocystic acne (**a**) extending to the forearms and thighs with syndactyly (**b**) in a 16-year-old boy with a germ-line *S252W FGFR2* mutation. (**a**) and (**b**) before and (**c**) 4 weeks after daily treatment with 40 mg isotretinoin. Reproduced from Chen et al. [87] with permission from Wiley, Hoboken, NJ, USA.

**Figure 7 biomedicines-09-01735-f007:**
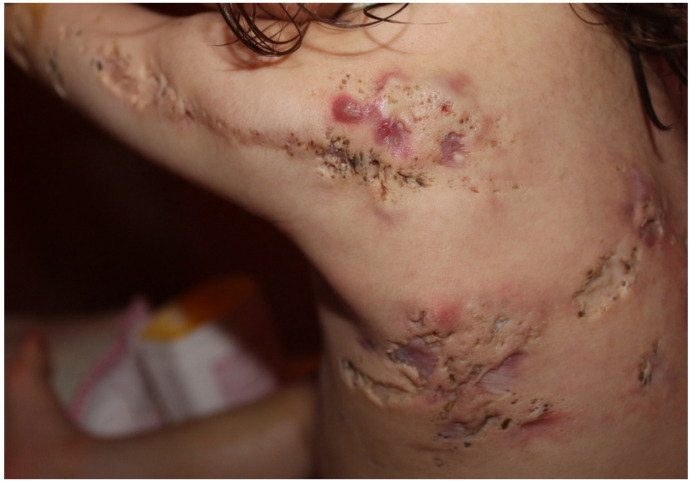
Nevus comedonicus. Multiple nodulocystic lesions, atrophic scars, fibrous tracts, and grouped comedone-like pits scattered unilaterally on the left side of the trunk, superior limb and the retro-auricular area of a 2 year and 7-month-old girl, born of a non-consanguineous marriage. Courtesy by Prof. Anca Chiriac, Department of Dermatology, Nicolina Medical Center, Apollonia University, Iași, Romania.

**Figure 8 biomedicines-09-01735-f008:**
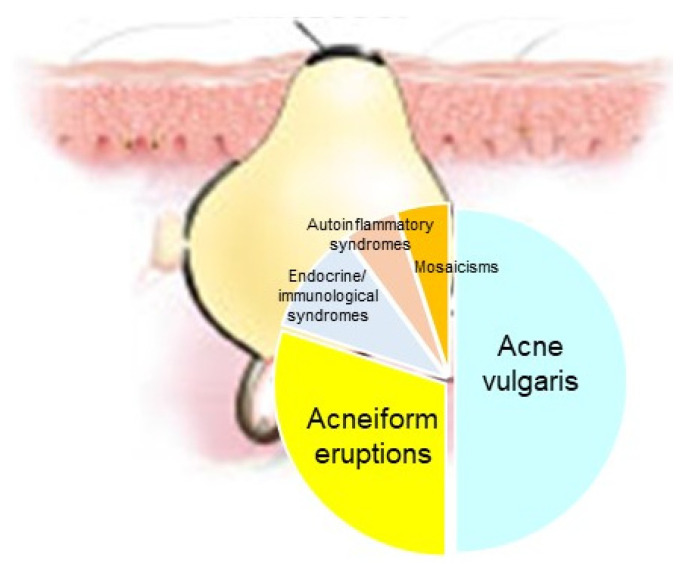
Acne represents a cardinal skin manifestation in several groups of diseases, such as acne vulgaris, acneiform eruptions (acneiform dernatoses), endocrine/immunological syndromes, autoinflammatory syndromes and certain mosaicisms.

**Table 1 biomedicines-09-01735-t001:** Syndromal acne.

Endocrinologic Syndromes	Autoinflammatory Syndromes
Congenital adrenal hyperplasia Polycystic ovary syndrome Seborrhea-acne-hirsutism-androgenetic alopecia syndrome Primary ovarian insufficiency	Pyogenic arthritis-pyoderma gangrenosum-acne syndrome Pyoderma gangrenosum-acne-suppurative hidradenitis syndrome Pyogenic arthritis-pyoderma gangrenosum-acne-suppurative hidradenitis syndrome Psoriatic arthritis-pyoderma gangrenosum-acne-suppurative hidradenitis syndrome Pustular psoriasis-arthritis-pyoderma gangrenosum-synovitis-acne-suppurative hidradenitis syndrome Pyoderma gangrenosum-acne-suppurative hidradenitis-ankylosing spondylitis syndrome Synovitis-acne-pustulosis-hyperostosis-osteitis syndrome

**Table 2 biomedicines-09-01735-t002:** Classification of SAHA syndrome.

Syndrome Variant	Characteristics
• Familial	Ethnic hyperandrogenism: common in southern Europe and Arab countries, usually no alteration of peripheral hormonal levels
• Ovarian	Most common cause of hyperandrogenism, often PCOS
• Adrenal	Anatomical or only functional adrenal hyperplasia
• Hyperprolactinemic	Clinical manifestations similar with those of adrenal SAHA, increased serum prolactin
• HAIRAN syndrome	Hyperandrogenism, insulin resistance, acanthosis nigricans, variant of SAHA syndrome with polyendocrinopathy

**Table 3 biomedicines-09-01735-t003:** Acneiform eruptions not being part of a systemic disease or syndrome.

Acne aestevalis Acne medicamentosa Drug-induced acne Steroid-induced acne Rosacea Perioral dermatitis Folliculitis, Gram-negative folliculitis Tinea barbae Demodicosis

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
