# Peer review of "Acne Syndromes and Mosaicism"

_biomedicines, 2021, doi:10.3390/biomedicines9111735_

Round 1
Reviewer 1 Report
An interesting narrative review exploring all the various acne-related syndromes and genetic issues; Although no new data is generated and no big improvement or new information is contained in the paper, I found it very readable and very useful to clinicians approaching syndromic and genetic causes of acne, so I think the paper will be publishable after revisions:
A materials and methods section, stating what keywords used in order to select the studies included in this review, as well as the databases searched would be a great addition to this study
Also an introduction section before talking about mosaicism, stating why you did perform this review and how it could be useful to researchers/clinicians would be very appreciated.
Subchapter 2 after" Other forms of treatment include oral combined
contraceptive medication for females with acne" you should add: "also photodinamic therapy has been proposed to manage inflammatory acne" and cite an article such as: doi: 10.23736/S0392-0488.19.06392-2.
Subchapter 2 after :"The depth and extent of acne scarring varies and can be improved by multiple procedures including subcision, punch excision,laser resurfacing, dermabrasion, and chemical peels" you should add: among these treatments, laser therapy seems the most effective to manage this aesthetic outcome" and cite an article such as: doi: 10.1007/s10103-020-03063-6.
Author Response
Reviewer 1
An interesting narrative review exploring all the various acne-related syndromes and genetic issues; Although no new data is generated and no big improvement or new information is contained in the paper, I found it very readable and very useful to clinicians approaching syndromic and genetic causes of acne, so I think the paper will be publishable after revisions:
The authors thank the reviewer for his/her kind comments.
A materials and methods section, stating what keywords used in order to select the studies included in this review, as well as the databases searched would be a great addition to this study
Our manuscript represents a review on the field but not a systematic review. However, we, indeed, performed a search, which is now included in the revised version of the manuscript.
Also an introduction section before talking about mosaicism, stating why you did perform this review and how it could be useful to researchers/clinicians would be very appreciated.
The requested introduction section has now been added to the manuscript.
Subchapter 2 after" Other forms of treatment include oral combined contraceptive medication for females with acne" you should add: "also photodinamic therapy has been proposed to manage inflammatory acne" and cite an article such as: doi: 10.23736/S0392-0488.19.06392-2.
Both the comment and the citation have been added.
Subchapter 2 after :"The depth and extent of acne scarring varies and can be improved by multiple procedures including subcision, punch excision,laser resurfacing, dermabrasion, and chemical peels" you should add: among these treatments, laser therapy seems the most effective to manage this aesthetic outcome" and cite an article such as: doi: 10.1007/s10103-020-03063-6.
Both the comment and the citation have been added.
Reviewer 2 Report
Thank you for the interesting review with a fresh view on the etiology of this dermatologic disease.
I have a couple of remarks/suggestions for improvement:
- ‘3. Acne mosaicism: Since family history of acne is a risk factor that predisposes an individual to develop the condition, genetics may play a pivotal role in the pathogenesis. Several theories have been proposed with regards to genetic causes, such as decreased CAG repeats,…’
→ decreased CAG repeats in what? (AR gene)
- ‘FGFR2 is the gene encoding FGFR2…’
→That doesn’t make sense, the fibroblast growth factor receptors comprise a family of related but individually distinct tyrosine kinase receptors. Please correct.
- Table 1: The distinction between ‘Systemic disorders and syndromes’ and ‘Autoinflammatory syndromes’ is arbitrary and the table does not contribute to clarity. My advice would be to abandon the table entirely.
- Table 2: very detailed and redundant
- Table 3: necessary information but disorderly, please adjust
- ‘7.1: More than 60 different mutations have been identified for the syndromic forms of craniosynostosis [137].’
→superfluous information in this context
- Fig. 7 is randomly added at the end without explanation or discussion: either add a paragraph with explanation or lose the figure

Author Response
Reviewer 2
Thank you for the interesting review with a fresh view on the etiology of this dermatologic disease.
The authors thank the reviewer for his/her kind comments.
I have a couple of remarks/suggestions for improvement:
‘3. Acne mosaicism: Since family history of acne is a risk factor that predisposes an individual to develop the condition, genetics may play a pivotal role in the pathogenesis. Several theories have been proposed with regards to genetic causes, such as decreased CAG repeats,…’
The correction of these sentences has been performed at the revised manuscript.
→ decreased CAG repeats in what? (AR gene)
The term “Androgen receptor gene” has been added, as AR.
‘FGFR2 is the gene encoding FGFR2…’
→That doesn’t make sense, the fibroblast growth factor receptors comprise a family of related but individually distinct tyrosine kinase receptors. Please correct.
We thank the reviewer for the attention to this erroneous statement, which has been respectively corrected.
Table 1: The distinction between ‘Systemic disorders and syndromes’ and ‘Autoinflammatory syndromes’ is arbitrary and the table does not contribute to clarity. My advice would be to abandon the table entirely.
We have substituted the term “Systemic disorders and syndromes” through “Endocrinologic syndromes” to add clarity to our classification and kept the table in our manuscript.
Table 2: very detailed and redundant
The SAHA syndrome is a disorder been described in its complete form by the authors group. The authors experience very often missing knowledge and understanding of physicians regarding this group of disorders, therefore, decided to include the table in the manuscript.
Table 3: necessary information but disorderly, please adjust
The initial used order was alphabetic, now we have ordered the diseases according to their etiologic background.
‘7.1: More than 60 different mutations have been identified for the syndromic forms of craniosynostosis [137].’
→superfluous information in this context
The sentence has been deleted, as proposed by the reviewer.
Fig. 7 is randomly added at the end without explanation or discussion: either add a paragraph with explanation or lose the figure
Fig. 7 shows a nevus comedonicus, the legend has been improved to provide understanding of the figure and the figure reference has been added to the text.
Round 2
Reviewer 1 Report
The paper improved after revisions. It is in my opinion publishable.
Reviewer 2 Report
Thank you for your answers and adjustments, I have no additional remarks.